# Biological Effects of Scattered Versus Scanned Proton Beams on Normal Tissues in Total Body Irradiated Mice: Survival, Genotoxicity, Oxidative Stress and Inflammation

**DOI:** 10.3390/antiox9121170

**Published:** 2020-11-24

**Authors:** Samia Chaouni, Alexandre Leduc, Frédéric Pouzoulet, Ludovic De Marzi, Frédérique Megnin-Chanet, Dinu Stefan, Jean-Louis Habrand, François Sichel, Carine Laurent

**Affiliations:** 1Cancer Centre François Baclesse, Normandy University, UNICAEN, UNIROUEN, ABTE-EA4651, 14076 Caen, France; samia.chaouni@unicaen.fr (S.C.); alexandre.leduc@unicaen.fr (A.L.); d.stefan@baclesse.unicancer.fr (D.S.); jl.habrand@baclesse.unicancer.fr (J.-L.H.); francois.sichel@unicaen.fr (F.S.); 2Translational Research Department, Experimental Radiotherapy Platform Institut Curie, PSL Research University, 91401 Orsay, France; Frederic.Pouzoulet@curie.fr; 3Radiation Oncology Department, Proton Therapy Centre, Centre University, Institut Curie, PSL Research University, 91898 Orsay, France; ludovic.demarzi@curie.fr; 4Laboratoire d’Imagerie Translationnelle en Oncologie, INSERM, Institut Curie, PSL Research University, University Paris Saclay, 91401 Orsay, France; 5INSERM U1196/CNRS UMR 9187, Institut Curie-Recherche, University Paris-Saclay, 91405 Orsay, France; Frederique.Megnin@curie.fr; 6Radiotherapy Department, Cancer Centre François Baclesse, 14000 Caen, France; 7SAPHYN/ARCHADE (Advanced Resource Centre for HADrontherapy in Europe), 14000 Caen, France

**Keywords:** proton therapy, double scattering, pencil beam scanning, side effects, healthy tissues, genotoxicity, oxidative stress, inflammation, mice

## Abstract

Side effects of proton therapy are poorly studied. Moreover, the differences in the method of dose delivery on normal tissues are not taken into account when proton beams are scanned instead of being scattered. We proposed here to study the effects of both modalities of proton beam delivery on blood; skin; lung and heart in a murine model. In that purpose; C57BL/6 mice were total body irradiated by 190.6 MeV proton beams either by Double Scattering (DS) or by Pencil Beam Scanning (PBS) in the plateau phase before the Bragg Peak. Mouse survival was evaluated. Blood and organs were removed three months after irradiation. Biomarkers of genotoxicity; oxidative stress and inflammation were measured. Proton irradiation was shown to increase lymphocyte micronucleus frequency; lung superoxide dismutase activity; erythrocyte and skin glutathione peroxidase activity; erythrocyte catalase activity; lung; heart and skin oxidized glutathione level; erythrocyte and lung lipid peroxidation and erythrocyte protein carbonylation even 3 months post-irradiation. When comparing both methods of proton beam delivery; mouse survival was not different. However, PBS significantly increased lymphocyte micronucleus frequency; erythrocyte glutathione peroxidase activity and heart oxidized glutathione level compared to DS. These results point out the necessity to take into account the way of delivering dose in PT as it could influence late side effects.

## 1. Introduction

In a large number of cancer cases, radiotherapy (RT) by photons is the main tool for curing cancers. Associated or not with surgery, this therapy often allows definitive control of the tumor [1,2]. However, about 15–20% of the patients treated by RT, prognosis is not improved [3] of RT go to failure. These failures depend on different parameters: (i) characteristics of the tumor (volume, cell kinetics, radiosensitivity, repair capacity), (ii) its environment (hypoxia, proximity of organs at risk) [4,5] and (iii) the radiation sensitivity of surrounding healthy tissues [6]. The challenge to ameliorate RT is to increase the anti-tumoral efficiency while decreasing side effects on healthy tissues. In this way, different tracks have been developed to improve the effectiveness of the treatment: ballistic, biomodulation and imaging in order to limit the dose to healthy tissues [7].

Conventional RT uses photons to irradiate cancer cells and is characterized by an exponential dose decrease. Thus, the energy is deposited mainly upstream and also downstream of the tumor in healthy tissues leading to possible toxicities at short and/or long term after RT. In contrast with photons, protons have the advantage to stop at a given depth (where the tumor is located), with reduced straggling and resulting in a significant reduction of integral dose to surrounding tissues. Proton therapy (PT) is considered as an alternative to conventional RT, especially in pediatric patients [8]. Nowadays, protons are frequently used instead of photons due to their ballistic properties which allow to treat tumors localized near organs at risk (OAR) without leading to toxicities in these OAR [9]. The proton path in tissues takes place in the form of a Bragg peak, so the majority of the energy is deposited at the end of the path. In this way, downstream healthy tissues should be spared [10,11,12]. PT thus allows the treatment of tumors whose location is critical such as pediatric [13,14], ocular [15], head and neck [16], breast [17], lung [18], and some urological cancers [19].

The state of knowledge is insufficient to answer to many questions concerning side effects of PT on healthy tissues. Indeed, available data concern mainly relative biological effectiveness (RBE) calculated from cell death. Furthermore, the access to beam line is difficult and the facilities are often inadequate for biological experiments [2,11]. As a consequence, few studies have been performed on possible differences in the biological effects of proton beam delivery methods. Studies on proton biological effects have been conducted most of the time in vitro on tumor cells and in much smaller proportion on normal cells [20,21] or by some meta-analysis studies [22] in comparison with the larger number of studies on conventional radiotherapy. Regarding in vivo experiments concerning effects of protons on healthy tissues, studies are rarer. Recently, Choi et al. demonstrated that, in comparison to photons, proton beam irradiation can induce a higher toxicity in the gastrointestinal tract [23]. Many other authors report specific radiobiological impact for clinical proton beams, or significant differences in the final range and linear energy transfer which can modulate biological effectiveness [10]. However, proton therapy is still based on the use of a generic RBE of 1.1 [24,25], which is applied to all treatments without any dependence of dose/fraction, position in the irradiated volume, beam energy or type of tissue.

In proton therapy, there are two main delivery techniques. One is called passive scattering and consists in producing a broad beam, the other one is called active scanning or PBS and consists in scanning several small pencil beams throughout the target volume [26,27]. However, there is a lack of data relative to biological effects of scattered versus scanned proton beams on normal tissues. On one hand, passive delivery leads to an overdose in healthy tissues in the lateral parts of the proximal layers [28], on the other hand, active scanning conducts to strong dose rate disparities in normal and tumor tissues. Indeed, dose rate is completely different between the proximal and the distal layers where dose rate can be up to dozens of Gy/s. In the same way, the duration of the fraction given by an active proton beam can go from some seconds in the distal zone compared to several minutes in the proximal layer. Intrafractionation is also different as the chosen dose will be given in one pulse for the most distal part of the tumor whereas it will be delivered in a large number of pulses for the most proximal part [29]. These properties of DS and PBS proton beams could thus lead to biological effects in surrounding healthy tissues.

Indeed, despite all the advances that have been made on this new therapy, toxicities to healthy tissues still exist. In addition to the findings of Choi et al. [23] about the relative toxicity induced by PT in mouse model, Mac Donald et al. showed that treatment by passive protons of the left breast cancer induces a percentage of dose non–negligible upon heart and left lung [30]. Otherwise, Hall et al. demonstrated that PT is only effective while using the PBS whose dose is 10 times lower than that used in RT by photons [31]. In contrast, the study performed by Gridley et al. on the comparison of PBS and DS treatments effects on human lung epithelial cells did not show any advantage of PT by PBS compared to DS [32]. Moreover, in patients treated for breast cancer by carbon therapy, Matsubara et al. have even showed that active scanning was not always superior to passive scattering [33]. All of these results showed the necessity to carry out additional studies to determine the toxic effects related to both proton delivery techniques and also to investigate more precisely the underlying mechanisms behind the development of these secondary effects after irradiation.

Ionizing radiations lead to different cellular responses. Indeed, photons and protons can interact with oxygen and water molecules present in cells producing reactive oxygen species (ROS). Among these species, two are free oxygen radicals: the hydroxyl radical (^•^OH) and the superoxide anion (O_2_^•−^) [3,34]. At low concentrations, ROS are involved in the modulation of cell function (differentiation, proliferation) and play a role in inflammatory reactions and as second messengers in signaling pathways. At higher concentrations, radicals can be harmful. In response to stress and in order to protect themselves, cells activate biochemical defense systems involved in the metabolism of ROS such as small molecules like HSP (Heat Shock Proteins) [35]. In the case where these defense systems are overwhelmed and/or ineffective, an imbalance occurs in favor of free radicals, then oxidative stress occurs. This leads to the disruption of cell signaling pathways, redox status control and/or multiple molecular damage [36]. Radicals attack biological macromolecules in the direct environment of their place of production. These radicals being very reactive and therefore having a short lifespan, all biological molecules having double bonds are likely to be affected. As a consequence, degradation products are generated (oxidized bases and cleavages of nuclear and mitochondrial DNA, products of lipid peroxidation, oxidized proteins, oxidized cholesterol). These breakdown products have a longer lifespan and some of them can cause gene activation or repression such as ATM, P53 [37].

Since it is essential to preserve enough ROS to ensure their function as secondary messengers, it is of major importance to control their concentration in order to avoid an excess inside cells and tissues. In order to counteract ROS accumulation, cells possess defense systems constituted of non-enzymatic molecules (vitamin E, ascorbic acid, glutathione) and enzymatic molecules (catalase, superoxide dismutase, glutathione peroxidase). Both of these systems play an essential role in protecting cells against radicals [7]. Interestingly, Baluchamy et al. conducted a study on mouse brain that aimed at evaluating the state of oxidative stress induced by protons. They demonstrated that PT exerts an indirect effect which results in an increase in the expression of some genes that up-regulate ROS [38]. Different other studies showed that exposure to radiations have an impact on oxidative stress in mouse intestine and colonic epithelial cells by functional dysregulation of mitochondria besides to an increase in oxidase activity [39] in addition to a loss of superoxide dismutase (SOD) and catalase (CAT) activity [40]. Moreover, Chi et al. demonstrated that X-ray irradiation induces an increase in lipid peroxidation (LPO) and in ascorbyl radical in mouse skin [41]. Barshishat-Kupper et al. showed that there is a protein oxidation in bone marrow, liver and lungs of C57BL/6J mice in response to irradiation [42,43,44]. Studying effects of radio-induced oxidative stress by targeting some biomarkers is necessary to understand different process leading to the development of some toxicities.

The purpose of this work was to assess the response of healthy tissues (skin, lung, heart and blood) after scattered or scanned proton beam irradiation. C57BL/6 mice were total body irradiated by DS or PBS at different doses in the plateau phase of the Bragg Peak and samples (blood, skin, heart, lungs) were collected 3 months later. Survival of mice was evaluated, DNA damage were quantified by micronucleus assay and antioxidant enzyme activities, LPO and protein carbonylation were assessed by biochemical techniques.

## 2. Materials and Methods

### 2.1. Reagents

RPMI and FBS were obtained from Lifesciences, NADPH from Roche (Mannheim, Germany), Legend plexTM and Human Inflammation Panel from BioLegend (San Diego, CA, USA). All other chemicals were purchased from Sigma-Aldrich (Saint-Louis, MO, USA).

### 2.2. Animals and Ethics Considerations

Experiments on C57BL/6 mice were approved by the ethics committee of the Institute Curie CEEA-IC #118 (authorization APAFiS# 27721-2020101612316744-v3 given by National Authority) in compliance with the international guidelines of the European Community (2010/63/UE). The study was approved by the research ethic committee. Mice were purchased from Charles River Laboratories (L’Arbresle, France). For this study, a number (n = 24 per group) of 10 weeks-old male C57BL/6 mice, weighing 22–27 g were housed at the RadeXP plateform of Curie Institute, maintained in a 12 h/12 h light-dark cycle at 22 °C with free access to food and water.

### 2.3. Irradiation

C57BL/6 mice were Total Body Irradiated (TBI) either by DS or PBS at different proton doses (0, 6, 7.5, 10, 12 and 14 Gy) in the plateau phase of the Bragg peak with an energy corresponding to 190.6 MeV. Irradiation was performed on C230 IBA accelerator at room temperature in the proton therapy center of Curie Institute (Orsay, France). The dose rate used was the standard rate in conventional treatment: 8 Gy in 15 s for PBS (32 Gy/min) and 8 Gy in 3.3 min for DS (2.4 Gy/min). Control mice were transported in the same conditions. Mice were anesthetized with 4% isoflurane before being placed under irradiator. Legs and tail were fixed with a soft plaster to prevent movements. Anesthesia (1.5–2% isoflurane) was kept in the mask throughout the irradiation. After irradiation, mice were housed in the animal house of the RadeXP platform and a veterinary follow-up was performed. Animals were sacrificed when a loss of more than 20% compared to the initial weight was observed. Blood and organs were collected 3 months later.

### 2.4. Skin, Lung, Heart & Blood Sample Collection

Three months after irradiation, animals were induced with 5% isoflurane and maintained under the mask with 1.5–2% isoflurane. Blood samples were collected by cardiac puncture and sent freshly to the laboratory. Then each organ of interest (heart, skin and lungs) was collected, stored at −80 °C and sent to the laboratory.

### 2.5. Separation of Blood Components

Fresh collected blood samples were centrifuged at 1000× *g* during 10 min at room temperature then separated into 3 compartments: the first one was plasma samples which were frozen at −80 °C and used for quantification of inflammatory cytokines, the second one was lymphocyte rings used for micronucleus assay and the last one was erythrocyte pellets aliquoted and frozen at −80 °C for oxidative stress evaluation.

### 2.6. Cytochalasin-Blocked Micronucleus Assay

Lymphocyte ring from each sample was put in 12 well plate containing RPMI 1640 culture medium (10% Fetal Bovine Serum or FCS, 1% penicillin/streptomycin) supplemented with 2% of phytohemagglutinin (PHA). Plates were then incubated during 21 h at 37 °C, 5% CO_2_ in humid atmosphere. After incubation, cytochalasin B was added to a final concentration of 3µg/mL in each culture well. After an additional 29 h incubation, the content of each well was transferred to a FACS tubes, centrifuged at 180× *g* during 5 min at room temperature. A volume of 2 mL of cold 75 mM KCl solution was added to the pellet dropwise. Then, a solution of cold Acetic acid/MetOH solution was added. After centrifugation, the content of each pellet was spread on slides as described previously by Chaouni et al. [7]. Analysis was performed using an automated scoring system Metafer (Metasystems, Altlussheim, Germany).

### 2.7. Preparation of Erythrocytes Lysates

Erythrocyte lysis was performed as described by Chaouni et al. [7]. Briefly, Erythrocytes were placed in a lysis buffer and lysed by thermal shocks. Samples were then centrifuged and supernatants obtained were aliquoted and stored at −80 °C.

### 2.8. Preparation of Tissue Homogenates

Collected tissues were placed in tubes on ice in the presence of phosphate buffer containing: KH_2_PO_4_ 10 mM, Na_2_HPO_4_ 40 mM, EDTA 0.1 M, pH = 7.5 and were then homogenized using ULTRA-TURRAX^®^ (Staufen, Germany). Lung and heart homogenates were prepared following an increasing speed rate: 8000, 9500, 13,500, 20,500 rpm. Each step lasted 3 min. Concerning skin, the first step was set at 13,500 rpm during 5 min followed by 5 min at 20,500 rpm and finally 5 min at 24,000 rpm. Homogenates were then centrifuged at 15,000× *g* during 10 min at 4 °C. The supernatant obtained were aliquoted and stored at −80 °C.

### 2.9. Protein Quantification

The amount of proteins was determined using a Bradford assay. Briefly, 5 µL of diluted samples were added to 250 µL of Bradford reagent. A standard curve of BSA (Bovine Serum Albumine) was performed at the same time. Absorbance was measured at 595 nm using microplate reader (FLUOstar OPTIMA, BMG, Labtech, Cary, NC, USA). Protein concentrations were calculated according to the standard curve equation.

### 2.10. Measurement of SOD activity

Total SOD activity was measured using the Flohé and Otting method [45]. Briefly, erythrocyte lysates and tissue homogenates were diluted in bicarbonate buffer containing 50 mM NaHCO_3_, 50 mM Na_2_CO_3_, 0.5 mM DTPA, 57.2 mM xanthine and 0.77 mM NBT (Nitrotetrazolium blue chloride). Samples were then placed in a 96-well plate. After an incubation at 30 °C during 10 min, 20 µL of xanthine oxidase was added. SOD activity was monitored following the reduction of NBT measured by spectrophotometer (SYNERGY H1) at 560 nm every 15 s during 1.5 min. One unit of SOD is defined as the amount of enzyme necessary for the disproportionation of 50% of the superoxide radicals. SOD activity was determined using the calculation below. Results were expressed in IU.mL^−1^ .mg^−1^ of proteins.
$$\text{SOD activity}=\frac{\left(\%\;\mathrm{inhibition}\times 50\right)}{\left(\mathrm{Vtot}\times \mathrm{Vs}\right)}$$


SOD: superoxide dismutase, Vtot: total volume, Vs: volume of diluted sample.

### 2.11. Measurement of CAT Activity

CAT activity was quantified as described by Chaouni et al. [7] in erythrocyte lysates and organ homogenates. Briefly, samples were diluted in a 50 mM potassium phosphate buffer (99:1, *v*/*v*) and 25 µL of diluted samples were placed in UV-star plates (Greiner Bio-One, Kremsmünster, Austria). The reaction was initiated by the addition of 225 µL of 30 mM hydrogen peroxide. The decrease in absorbance was measured at 240 nm during 1 min by a microplate reader Synergie H1 (Biotech Labs, Rockville, MD, USA). CAT activity was calculated according to slope values from the standard curve (purified liver CAT) and results were expressed as nmol of consumed hydrogen peroxide per min per mg of proteins.

### 2.12. Measurement of Glutathione Peroxidase (GPx) Activity

GPx activity was quantified as described by Chaouni et al. [7] in erythrocyte lysates and organ homogenates. Briefly, samples were diluted in a buffer containing 125 mM potassium phosphate buffer (Na_2_HPO_4_/NaH_2_PO_4_, pH 7), 12.5 mM EDTA, 50 mM KCN, 5 mM NADPH, 5 mM reduced glutathione and 0.25 IU of glutathione reductase and diluted samples were place in 96-well plates for 15 min at 30 °C. Reaction was initiated by the addition of tert-butyl hydroperoxide (t-BuOOH) at 250 µM. The decrease in absorbance was measured at 340 nm for 2.5 min by a microplate reader Synergie H1 (Biotech Labs, Rockville, MD, USA). GPx activity was calculated according to the equation below and results were expressed as nmol of oxidized glutathione per min per mg of proteins.
$$\text{GPx activity}=2\times \frac{\frac{\left|\mathrm{slope}\right|\;({\mathrm{AU}.\min}^{-1})}{\xi_{\mathrm{NADPH}}\left({\mathrm{cm}}^{-1}\right)\times l\;(\mathrm{cm})}\times \frac{\mathrm{Vf}\;(\mathrm{mL})}{\mathrm{Vs}\;(\mathrm{mL})}\times \text{sample dilution}}{\mathrm{protein}\;\mathrm{concentration}\;(\mathrm{mg}/\mathrm{mL})}$$


AU: Absorbance Unit, ξ: molar extinction coefficient for NADPH (Nicotinamide Adenine Dinucleotide Phosphate) at 340 nm (0.00622 µm^−1^.cm^−1^), l: optical path length, Vf: final volume per well, Vs: volume of diluted sample.

### 2.13. Quantification of Oxidized Glutathione (GSSG)

A quantification kit was used to measure GSSG according to the manufacturer’s recommendations. Briefly, GSSG was reduced in GSH by the addition of NADPH and glutathione reductase. GSH then reacts with 5,5′-dithiobis-2-nitrobenzoic acid to form a product which can be measured by spectrophotometry (412 nm). A masking reagent was added to trap the initial GSH in order to measure only oxidized glutathione. Level of GSSG was measured by a microplate reader Synergie H1 (Biotech Labs, Rockville, MD, USA) and calculated from the standard curve. Results were expressed in µmol of GSSG per mg of proteins.

### 2.14. Measurement of LPO

LPO was evaluated as described by Chaouni et al. [7] in erythrocyte lysates and organ homogenates using PeroxiDetect Kit according to the manufacturer’s recommendations. Peroxides react with Fe^2+^ ions and produce Fe^3+^ ions in the same proportion of hydroperoxides present in samples. Then, Fe^3+^ ions react with xylenol orange (3,3′-bis[*N*,*N*-bis(carboxymethyl)aminomethyl]-o-cresolsulfonephtalein, sodium salt) to form a colored compound detected by spectrophotometry (570 nm). The amount of lipid hydroperoxides was measured by a microplate reader Synergie H1 (Biotech Labs, Rockville, MD, USA) and calculated from the standard curve of t-BuOOH. Results are expressed as nmol of peroxides per mg of proteins.

### 2.15. Measurement of Protein Carbonylation

Protein carbonylation was evaluated as described by Chaouni et al. [7] in erythrocyte lysates and organ homogenates using a protein carbonyl content assay kit according to the manufacturer’s recommendations. Samples were derived in dinitrophenyl (DNP) hydrazone adducts by 2,4-dinitrophenylhydrazine (DNPH). Trichloroacetic acid was added to precipitate proteins. After an acetone washing step to remove excess DNPH and retain only proteins, a centrifugation was performed and pellets were suspended in a guanidine solution (6 M). The amount of protein carbonyls was measured at 375 nm by a microplate reader Synergie H1 (Biotech Labs, USA) and calculated from equation below. Results are expressed as nmol of protein carbonyls per mg of proteins.
Protein carbonyls (nmol carbonyls/mg protein) = (C/P) × 1000 × D


C: amount of carbonyls in sample wells (nmol/well), P: amount of proteins from standard wells, D: dilution factor of samples, 1000: conversion factor (µg to mg).

### 2.16. Cytokine Quantification in Plasma

A panel of 13 cytokines (IL-1ß, IFN-α2, IFN-y, TNF-α, MCP-1, IL-6, IL-8, IL-10, IL-12p70, IL-17A, IL-18, IL-23, IL-33) was measured in plasma by using Legendplex TM Human Inflammation Panel as previously described by Chaouni et al. [7]. Analysis was perfomed by flow cytometry using a CytoFlex flow cytometer (Beckman Coulter, Brea, CA, USA) and data were analyzed by the data analysis software LEGENDplex™ version 8, (BioLegend, San Diego, CA, USA).

### 2.17. Statistical Analysis

Results are presented as mean values ± standard error of the mean (SEM). * for *p* < 0.05, ** for *p* < 0.005 and *** for *p* < 0.0001 for irradiated versus unirradiated samples and # for *p* < 0.05, ## for *p* < 0.005 and ### for *p* < 0.0001 for DS versus PBS (two-way ANOVA and Mann–Whitney test). Experiments were performed in triplicates.

## 3. Results

### 3.1. Evaluation of Survival of Mice

Mouse survival curves showed significant decrease after irradiation at doses from 8 Gy (data not shown). Chosen irradiation dose for next experiments was the sublethal dose of 7.5 Gy (Figure 1). No significant difference was observed between the two types of proton delivery (Logrank test, *p* = 0.5542, with n = 24 for PBS and n = 25 for DS).

### 3.2. Lymphocyte Micronucleus Frequency and Distribution

In order to assess chromosome breaks and repairs induced by irradiation, we performed a CBMN assay to study genotoxicity (Figure 2). Our results showed that both irradiation delivery techniques induced an increase in lymphocyte micronucleus frequency with a significant 4.3- and a non-significant 2.6-fold increase after PBS and after DS, respectively, in comparison to non-irradiated mice (Figure 2a). When comparing PBS and DS, there was a significant 1.6-fold increase in micronucleus frequency in PBS compared to DS irradiated mice. In addition, DS irradiation led to a significant higher level of undamaged lymphocytes whereas PBS irradiation seemed to induce a higher rate of strongly damaged lymphocytes in comparison to DS irradiation (Figure 2b).

### 3.3. Evaluation of Antioxidant Enzymes Activities and Oxidized Glutathione in Erythrocytes Lysates and Tissue Homogenates

To evaluate the oxidative state in healthy tissues after irradiation, we analyzed the activity of the main 3 enzymes: SOD, CAT and GPx in addition to glutathione oxidized (GSSG) in erythrocyte lysates and collected organs (lung, heart and skin).

Concerning SOD activity (Table 1), our results showed that, irradiation induces an increase in SOD activity in all collected samples from irradiated mice compared to non-irradiated ones, except in hearts from mice exposed to DS. This increase was significant in lungs in both groups. Regarding other organs, the same trend to increase was observed after DS irradiation with respectively 1.2-fold change after PBS vs. 1.2-fold change after DS in erythrocytes lysates and 1.5-fold change after PBS irradiation vs. 1.6-fold change after DS irradiation in skin samples. Concerning heart samples, the same trend to increase was observed after PBS irradiation however the contrary was observed after DS irradiation, but this was not significant.

Concerning GPx activity (Table 2), our results showed a significant increase in erythrocyte and skin homogenates after irradiation. There was a 1.3-fold higher GPx activity in erythrocyte lysates after PBS than after DS. There was a same trend in other organs to more GPx activity after PBS than after DS: 1.1-fold and 1.3-fold higher GPx activity in lungs and in skin, respectively. However, the contrary was observed in heart samples with a 0.82-fold change after PBS vs. DS irradiation.

CAT activity (Table 3) was globally increased after irradiation, but significantly only for erythrocytes with PBS mode. In heart samples, CAT activity seemed to be unchanged after DS and even decreased after PBS. When comparing PBS and DS irradiation, there was a significant variation only for erythrocytes with a 1.4-fold increase after PBS compared to DS. A non-significant strong increase can be noted concerning lung CAT activity with a 1.6-fold higher value after DS compared to PBS. However, it should be mentioned that CAT activity values for heart and lungs were very low making interpretation quite difficult.

Finally, results concerning oxidized glutathione (Table 4) showed a high significant increase in GSSG level in lungs and heart after PBS irradiation and in skin after DS irradiation. There were significant changes in GSSG between PBS and DS irradiation. On one hand, there was a 1.5-fold increase in heart samples after PBS compared to DS, one the other hand, there was a significant 1.6-fold increase in skin samples after DS compared to PBS.

### 3.4. LPO in Erythrocytes Lysates and Tissue Homogenates

LPO was quantified in order to assess the oxidative state of lipids. Our results showed (Table 5) a significant increase in LPO rate with PBS irradiation compared to controls in erythrocytes lysates. The same trend to increase was observed in lung but was significant only after. For all samples, no remarkable variations were observed between PBS vs. DS. However, trends were observed with a higher LPO content in erythrocyte lysates and heart homogenates after PBS than after DS whereas the contrary was observed in lungs. Concerning skin samples, LPO level was below the detection threshold.

### 3.5. Protein Carbonylation in Erythrocytes Lysates and Tissue Homogenates

Protein carbonylation was measured (Table 6) and revealed a significant increase in carbonyl level in erythrocytes lysates after PBS or DS irradiation. No significant changes were observed between both types of proton delivery. However, there was a trend to a lower carbonyl level after DS than after PBS irradiation. Concerning skin samples, carbonyl level was below the detection threshold.

### 3.6. Quantification of Plasma Inflammatory Cytokines

After measurement of 13 cytokines level in plasma samples, our results showed that irradiation induced a non-significant increase in IL-1α level with a 1.4- and a 1.3-fold increase after PBS and DS, respectively, compared to unirradiated samples (Figure 3a). In contrast, a non–significant decrease of TNF-α level was observed for both groups with a 1.1- and a 1.5-fold decrease after PBS and DS, respectively, compared to unirradiated samples (Figure 3b). Concerning MCP-1 and IL-1β, no differences were observed after irradiation (data not shown). The levels of IL-inter6/10/12p70/17A/23/33 and IFN-γ2/y were below the detection threshold (data not shown).

## 4. Discussion

Few data on side effects of proton therapy are available and they are not sufficient to answer to many questions related to the biological effects of proton exposure on healthy tissues. Moreover, proton delivery can be performed by two different techniques, passive or active, leading to differences in terms of dose, but also dose rate and fractionation. The present study aimed at evaluating normal tissue response (blood, lung, heart and skin) 3 months after TBI either by PBS or DS in C57BL/6 mouse model.

First of all, survival of mice was assessed at various doses and allowed to choose the sublethal dose of 7.5 Gy for next assays. At this dose, there was no significant difference between survival curves after PBS or after DS delivery.

CBMN assay allowed to evaluate DNA damage following exposure to radiations. Our results showed that whole body proton irradiation induced a high lymphocyte micronucleus frequency. Chang et al. also demonstrated that whole body proton irradiation of C57BL/6J mice led to an increase in DNA damage in hematopoietic stem cells 22 weeks after exposure [46]. Previous studies have also shown an increase in DNA damage after X-ray TBI exposure of C57BL/6 mice with an increased expression of γ-H2AX [47,48]. In our study, micronucleus frequency was significantly higher in PBS than in DS irradiated mice. This observation can be correlated to the stronger rate of highly damaged lymphocytes in PBS than in DS irradiated mice. This result suggests that PBS irradiation leads to more residual damage on lymphocytes in comparison to DS. This is in agreement with the study performed by Michaelidesova et al. on normal human fibroblasts showing that PBS proton beam delivery was more efficient to produce double-strand breaks [49]. The same authors showed there were no differences in micronucleus frequency or γ-H2AX foci in cancer cell lines (medulloblastoma) at early times after PBS or DS proton irradiation [50]. It could be interesting to evaluate in vitro the ability of both delivery modes to induce long term genotoxicity in primary normal cells.

One of the most known mechanisms of radiation effect is to cause an imbalance in redox homeostasis by the production of ROS thus generating an oxidative stress. To restore cell homeostasis, different molecules involved in the antioxidant defenses are activated to avoid ROS accumulation [38]. SOD, CAT, GPx and GSSG are molecules at the front line in the detoxification chain. It is of major importance to measure their level in order to evaluate the antioxidant capacity in erythrocyte lysates and organ homogenates. Our results concerning antioxidant activities are changing according to the radiation dose and the organ. The most relevant variations after irradiation compared to unirradiated mice were an increase in SOD activity in lungs, in GPx activity in erythrocytes and skin, in CAT activity in erythrocytes and in GSSG level in lungs, heart and skin. In the same way, previous studies concerning total-body proton irradiation of mice led to a differential modulation of oxidative stress gene expression in liver with an early increase in Prdx6 and Sod3, mainly, whereas other genes were common to photon irradiation [21] and to a late increase in ROS production and NOX4 transcription in hematopoietic stem cells from irradiated mice 22 weeks after exposure [46]. Liao et al. have also shown that proton irradiation induces an overexpression of CAT which has a neuroprotective effect on hippocampal cell proliferation [51]. Contrary to results observed after proton TBI, 5 months after X-ray whole thorax C57BL/6 irradiation, Pan et al. observed a significant decrease in SOD, CAT and GPx activities [52]. However, results are contradictory with X-rays as other studies demonstrated that there is an increase in CAT activity in chicken tissues: kidney, brain, muscle, and liver [53], an increase in SOD activity but a decrease in GPx in TBI irradiated C3H/HeN mice [54]. ROS were also shown to increase in lungs after X-ray TBI [48]. When looking at the differences between PBS and DS techniques, the only significant changes that were observed were a decrease in CAT and GPx activity in erythrocytes and in GSSG level in heart after DS compared to PBS whereas an increase in GSSG level was observed after DS compared to PBS in skin. Globally, GPx activity is higher after PBS than after DS. It would be interesting to assess glutathione reductase activity which is part of the cycle regenerating GSSG into GSH. These results tend to show that active scanning and passive delivery result in completely different profiles of antioxidant capacity.

In order to evaluate damage to lipids and proteins, LPO and protein carbonylation were measured. After irradiation, the level of lipid hydroperoxides was significantly increased in erythrocytes of mice exposed to PBS and in lungs of mice exposed to DS compared to unirradiated samples. Baluchamy et al. have also shown an increase in LPO in mouse brain after proton irradiation [55]. In the same way, after photon irradiation, Iizawa et al. have shown an increase in LPO level in skin homogenates of C57BL/6 mice 18 h post irradiation [56]. This was confirmed by Chi et al. after a high dose of X-rays [41]. Regarding the difference between PBS and DS, no significant changes were observed in LPO in the different organs. Concerning protein carbonyls, their level was significantly increased in erythrocytes whatever the proton delivery technique. Barshishat-Kupper et al. used mass spectrometry technic and identified 7 carbonylated proteins in C57BL/6J lung samples 24 h post thoracic photon irradiation [43]. In addition, carbonylation of four proteins were increased in bone marrow of C57BL/6J mice exposed to 7.5Gy 60Co TBI [44]. As for LPO, protein carbonylation was not significantly changed between both proton delivery modes. However, a trend to a higher level of carbonyls was observed after PBS compared to DS. In this way, active scanning seems to lead to higher damage to macromolecules and higher antioxidant capacity depending on the organ considered. For LPO and protein carbonylation, the use of more sensitive techniques would be interesting as well as the measurement of adducts to DNA. Indeed, MDA is a well-known LPO product and can react with 2′-deoxyguanosine, 2′-deoxyadenosine, and 2′-deoxycytidine (for review, [57]). These adducts can be measured by ultra-high performance liquid chromatography-electrospray ionization analytical method coupled to mass spectrometry in the tandem mode [58]. In the same manner, identification of specific proteins being carbonylated after proton irradiation should be interesting by using an OxyBlot Oxidized Protein Detection Kit (Millipore) [43].

Inflammatory cytokine IL-1α was not significantly increased after proton irradiation. Surprisingly, TNF-α was decreased after proton irradiation, even if not significantly. Indeed, inflammatory cytokines were already shown to increase in mouse skin after irradiation [59]. However, Budagov et al. did not see differences in plasmatic inflammatory cytokines in irradiated mice [60]. Moreover, Warren and Moore showed that TNF administration in mice could protect hematopoietic stem cells from radiation effects, which was not the case with IL-1 administration [61]. Therefore, the trend to a decrease in TNF-α and to an increase in IL-1α after proton irradiation could demonstrate that PT leads to toxicities, even 3 months after irradiation. Concerning the different techniques of proton delivery, a same trend was observed in our experiments for both cytokines: PBS led to a higher level than DS proton delivery. Gridley et al. measured TNF-α in supernatants of human lung epithelial cells exposed either to active or passive proton beams but could not detect it, even in supernatants of unirradiated cells [32].

In summary, our findings indicate that proton radiation exposure has an effect on antioxidant enzymes activities, glutathione level and lipid and protein oxidation even long time after irradiation. This suggests that the oxidative state remains changed long time after irradiation coming either from the irradiation itself or from waves of oxidative stress resulting, for example, from inflammatory phenomena. Concerning the two types of proton delivery, our results let us hypothesize that PBS, even if survival of mice was the same as for DS, leads to different response according to the organs considered in terms of genotoxicity, antioxidant capacity, LPO, protein carbonylation and inflammatory cytokines. In general, PBS led to trends to an increase in micronucleus frequency, in GPx activity, in protein carbonyls and in IL-1α and TNF-α.

## 5. Conclusions

Few studies have been carried out on biological effects of proton therapy on healthy tissues. In addition, few scientific projects have looked at the comparison of effects of the difference in dose delivery to healthy tissues by the two types of beams used in proton therapy: scanned and scattered. These side effects could be linked to oxidative phenomena occurring after exposure to ionizing radiations. The response of lungs and heart in terms of long-term oxidative stress is an essential path that needs to be explored because these organs are considered as OAR and skin is also a crucial organ as its adverse effects are among the most common after RT: more than 50% for acute and up to 50% for late side effects concerning breast cancer [62]. The aim of this study was to assess the relative toxicity of the two modes of proton delivery PBS vs. DS on healthy tissues. Our different findings allowed us to obtain some responses about the toxic effects of PT by the identification of some specific biomarkers. Possible correlations with clinics, like radionecrosis in pediatric patients treated by PT for brain tumors [63], and physics data, like LET (linear energy transfer) maps, should be very interesting. Moreover, this study pointed out the necessity to take into account the specificity of dose delivery but also of each heathy tissue crossed by the beams.

## Figures and Tables

**Figure 1 antioxidants-09-01170-f001:**
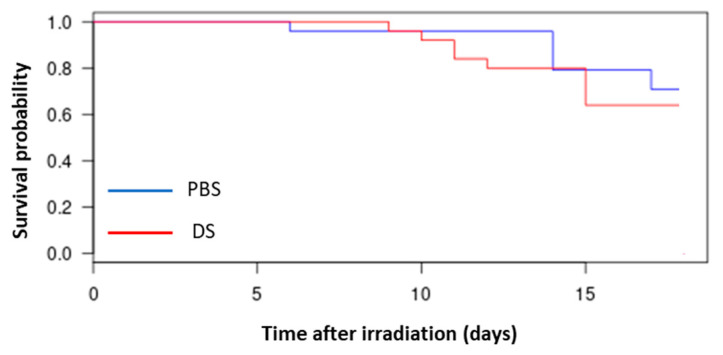
Survival of mice after 7.5 Gy proton TBI irradiation by PBS (Pencil Beam Scanning) or DS (Double Scattering) in the plateau phase of the Bragg peak. n = 24 for PBS and n = 25 for DS.

**Figure 2 antioxidants-09-01170-f002:**
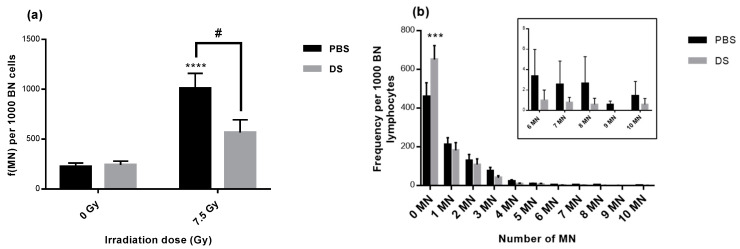
Lymphocyte micronucleus frequency (**a**) and distribution (**b**) at 7.5 Gy in mice irradiated by PBS or DS proton beams in the plateau phase of the Bragg peak. MN: micronucleus, BN: binucleated. Data are depicted as mean values ± SEM. *** for *p* < 0.0005 and **** for *p* < 0.0001 for irradiated samples compared to non-irradiated ones and # for *p* < 0.05 for DS compared to PBS. Each experiment was performed in triplicates.

**Figure 3 antioxidants-09-01170-f003:**
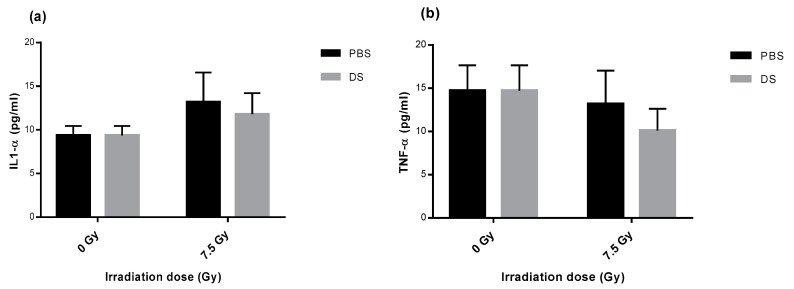
IL-1α (**a**) and TNF-α (**b**) concentration in plasma of mice irradiated by PBS or DS proton beams in the plateau phase of the Bragg peak. Data are depicted as mean values ± SEM. Each experiment was performed in triplicates.

**Table 1 antioxidants-09-01170-t001:** SOD activity in erythrocytes, lung, heart and skin of mice irradiated by PBS or DS proton beams in the plateau phase of the Bragg peak.

		Irradiation Group				
	Organ	0 Gy	7.5 Gy PBS	% Increase ↑ % Decrease ↓	7.5 Gy DS	% Increase ↑ % Decrease ↓
SOD activity (U/mg of proteins)	Erythrocytes	27.29 ± 2.20	33.35 ± 2.53	22.2% ↑	33.68 ± 5.14	23.41% ↑
	Lung	58.51 ± 3.89	103.14 ± 14.81	76.27% ↑ *	119.96 ± 23.00	105.02% ↑ **
	Heart	87.11 ± 8.07	96.59 ± 10.45	10.88% ↑	61.00 ± 8.51	29.97% ↓
	Skin	125.61 ± 13.61	192.46 ± 26.59	53.22% ↑	201.45 ± 27.46	60.37% ↑

Percentages of increase of decrease are PBS- or DS-irradiated versus unirradiated. Data are depicted as mean values ± SEM. * for *p* < 0.05 and ** for *p* < 0.005 for irradiated samples compared to non-irradiated ones. Each experiment was performed in triplicates.

**Table 2 antioxidants-09-01170-t002:** GPx activity in erythrocytes, lung, heart and skin of mice irradiated by PBS or DS proton beams in the plateau phase of the Bragg peak.

		Irradiation Group				
	Organ	0 Gy	7.5 Gy PBS	% Increase ↑ % Decrease ↓	7.5 Gy DS	% Increase ↑ % Decrease ↓
GPx activity (nmol of GSSG/min/mg proteins)	Erythrocytes	178.46 ± 5.66	261.73 ± 13.49	46.66% ↑ *** ###	204.06 ± 12.13	14.34% ↑ ###
	Lung	1404.86 ± 118.57	1755.60 ± 224.37	24.96% ↑	1574.60 ± 111.07	12.08% ↑
	Heart	699.81 ± 60.83	721.15 ± 56.02	3.04% ↑	878.41 ± 76.2	25.52% ↑
	Skin	120.41 ± 16.24	194.12 ± 23.85	61.21% ↑ *	146.32 ± 4.16	21.51% ↑

Percentages of increase of decrease are PBS- or DS-irradiated versus unirradiated. Data are depicted as mean values ± SEM. * for *p* < 0.05 and *** for *p* < 0.0001 for irradiated samples compared to non-irradiated ones and ### for *p* < 0.0001 for DS compared to PBS. Each experiment was performed in triplicates.

**Table 3 antioxidants-09-01170-t003:** CAT activity in erythrocytes, lung, heart and skin of mice irradiated by PBS or DS proton beams in the plateau phase of the Bragg peak.

		Irradiation Group				
	Organ	0 Gy	7.5 Gy PBS	% Increase ↑ % Decrease ↓	7.5 Gy DS	% Increase ↑ % Decrease ↓
CAT activity (U/mg proteins)	Erythrocytes	3.50 ± 0.15	4.44 ± 0.36	25.42% ↑ *** ###	3.07 ± 0.13	12.28% ↑ ###
	Skin	2.79 ± 0.35	3.62 ± 0.40	29.74% ↑	3.63 ± 0.65	30.10% ↑
	Heart	0.12 ± 0.01	0.09 ± 0.002	25.00% ↓	0.12 ± 0.003	No change
	Lung	0.24 ± 0.04	0.25 ± 0.05	4.16% ↑	0.35 ± 0.06	66.66% ↑

Percentages of increase of decrease are PBS- or DS-irradiated versus unirradiated. Data are depicted as mean values ± SEM. *** for *p* < 0.0001 for irradiated samples compared to non-irradiated ones and ### for *p* < 0.0001 for DS compared to PBS. Each experiment was performed in triplicates.

**Table 4 antioxidants-09-01170-t004:** GSSG in erythrocytes, lung, heart and skin of mice irradiated by PBS or DS proton beams in the plateau phase of the Bragg peak.

		Irradiation Group				
	Organ	0 Gy	7.5 Gy PBS	% Increase ↑ % Decrease ↓	7.5 Gy DS	% Increase ↑ % Decrease ↓
GSSG (µmol/mg of proteins)	Erythrocytes	12.98 ± 0.42	12.45 ± 1.13	4.08% ↓	13.35 ± 1.33	2.85% ↑
	Lung	9.18 ± 0.95	16.63 ± 2.95	81.15% ↑ **	13.28 ± 1.63	44.66% ↑
	Heart	9.80 ± 1.04	18.96 ± 1.84	93.46% ↑ *** #	12.68 ± 1.90	29.38% ↑ #
	Skin	8.64 ± 0.74	8.99 ± 0.92	4.05% ↑ #	14.22 ± 2.73	7.03% ↑ *** #

Percentages of increase of decrease are PBS- or DS-irradiated versus unirradiated. Data are depicted as mean values ± SEM. ** for *p* < 0.005 and *** for *p* < 0.0001 for irradiated samples compared to non-irradiated ones and # for *p* < 0.05 for DS compared to PBS. Each experiment was performed in triplicates.

**Table 5 antioxidants-09-01170-t005:** LPO in erythrocytes, lung, heart and skin of mice irradiated by PBS or DS proton beams in the plateau phase of the Bragg peak.

		Irradiation Group				
	Organ	0 Gy	7.5 Gy PBS	% Increase ↑ % Decrease ↓	7.5 Gy DS	% Increase ↑ % Decrease ↓
LPO (nmol/mg of proteins)	Erythrocytes	57.12 ± 4.46	89.93 ± 9.63	57.44% ↑ ***	68.76 ± 5.35	20.37% ↑
	Lung	11.87 ± 1.43	16.12 ± 2.36	35.80% ↑	21.82 ± 3.77	83.82% ↑ **
	Heart	18.87 ± 1.26	22.74 ± 5.84	20.50% ↑	21.21 ± 0.91	12.40% ↑
	Skin	below detection limit				

Percentages of increase of decrease are PBS- or DS-irradiated versus unirradiated. Data are depicted as mean values ± SEM. ** for *p* < 0.005 and *** for *p* < 0.0001 for irradiated samples compared to non-irradiated ones. Each experiment was performed in triplicates.

**Table 6 antioxidants-09-01170-t006:** Carbonyls in erythrocytes, lung, heart and skin of mice irradiated by PBS or DS proton beams in the plateau phase of the Bragg peak.

		Irradiation Group				
	Organ	0 Gy	7.5 Gy PBS	% Increase ↑ % Decrease ↓	7.5 Gy DS	% Increase ↑ % Decrease ↓
Carbonyls (nmol/mg of proteins)	Erythrocytes	202.62 ± 5.72	273.36 ± 8.58	34.91% ↑ ***	234.85 ± 7.06	15.90% ↑ *
	Lung	11.92 ± 0.93	11.25 ± 1.46	5.62% ↑	10.73 ± 1.08	9.98% ↓
	Heart	7.04 ± 0.47	7.35 ± 1.04	4.40% ↑	6.53 ± 0.88	7.24% ↓
	Skin	below detection limit				

Percentages of increase of decrease are PBS- or DS-irradiated versus unirradiated. Data are depicted as mean values ± SEM. * for *p* < 0.05 and *** for *p* < 0.0001 for irradiated samples compared to non-irradiated ones. Each experiment was performed in triplicates.

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
