# Peer review of "Biological Effects of Scattered Versus Scanned Proton Beams on Normal Tissues in Total Body Irradiated Mice: Survival, Genotoxicity, Oxidative Stress and Inflammation"

_antioxidants, 2020, doi:10.3390/antiox9121170_

Round 1
Reviewer 1 Report
Chaouni et al. studied the effect of protontherapy on normal cells and tissues depending on two types of proton delivery. The topics covered are very relevant from a practical point of view and add value to the very few studies published in this field. They show the role of the proton delivery method in inducing side effects that appear later. The publication requires clarification of some issues.
Regarding the tabular presentation of the results. Please indicate precisely the statistically significant differences. Eg. in Table 2, the characters "###" are placed next to DS (up arrow), while these marks should be at PBS, because it was after PBS application that a significant increase in GPx compared to DS was observed.
Authors should also remove unnecessary characters from legends, not used in tables, eg "#" characters in the legend to Table 1 are redundant.
The authors should extend the description of the methodology for examining the parameters of the oxidative state.
line 72: explain abbreviation; RBE
Author Response
First, we would like to thank this reviewer for helpful comments.
Indeed, it is unclear concerning statistical differences between DS and PBS. # symbols are used to compare the two types of delivery so we decided to put the # symbols in both columns PBS and DS to be sure that the reader understands that the difference concerns the delivery mode in Tables 2 and 4.
As required, we deleted the unnecessary characters concerning statistics in the legends of all figures and tables.
We described more precisely the methods we used to measure:
- Catalase activity:
“CAT activity was quantified as described by Chaouni et al. [7] in erythrocyte lysates and organ homogenates. Briefly, samples were diluted in a 50 mM potassium phosphate buffer (99:1, v:v) and 25 µL of diluted samples were placed in UV-star plates (Greiner Bio-One, Kremsmünster, Austria). The reaction was initiated by the addition of 225 µL of 30 mM hydrogen peroxide. The decrease in absorbance was measured at 240 nm during 1 min by a microplate reader Synergie H1 (Biotech Labs, USA). CAT activity was calculated according to slope values from the standard curve (purified liver CAT) and results were expressed as nmol of consumed hydrogen peroxide per min per mg of proteins.”
- GPx activity:
“GPx activity was quantified as described by Chaouni et al. [7] in erythrocyte lysates and organ homogenates. Briefly, samples were diluted in a buffer containing 125 mM potassium phosphate buffer (Na2HPO4/NaH2PO4, pH 7), 12.5 mM EDTA, 50 mM KCN, 5 mM NADPH, 5 mM reduced glutathione and 0.25 IU of glutathione reductase and diluted samples were place in 96-well plates for 15 min at 30°C. Reaction was initiated by the addition of tert-butyl hydroperoxide (t-BuOOH) at 250 µM. The decrease in absorbance was measured at 340 nm for 2.5 min by a microplate reader Synergie H1 (Biotech Labs, USA). GPx activity was calculated according to the equation below and results were expressed as nmol of oxidized glutathione per min per mg of proteins.
|
AU: Absorbance Unit, x: molar extinction coefficient for NADPH (Nicotinamide Adenine Dinucleotide Phosphate) at 340 nm (0.00622 µm–1.cm–1), l: optical path length, Vf: final volume per well, Vs: volume of diluted sample.”
- GSSG level:
“A quantification kit was used to measure GSSG according to the manufacturer’s recommendations. Briefly, GSSG was reduced in GSH by the addition of NADPH and glutathione reductase. GSH then reacts with 5,5’-dithiobis-2-nitrobenzoic acid to form a product which can be measured by spectrophotometry (412 nm). A masking reagent was added to trap the initial GSH in order to measure only oxidized glutathione. Level of GSSG was measured by a microplate reader Synergie H1 (Biotech Labs, USA) and calculated from the standard curve. Results were expressed in µmol of GSSG per mg of proteins.”
- LPO measurement:
“LPO was evaluated as described by Chaouni et al. [7] in erythrocyte lysates and organ homogenates using PeroxiDetect Kit according to the manufacturer’s recommendations. Peroxides react with Fe2+ ions and produce Fe3+ ions in the same proportion of hydroperoxides present in samples. Then, Fe3+ ions react with xylenol orange (3,3′-bis[N,N-bis(carboxymethyl)aminomethyl]-o-cresolsulfonephtalein, sodium salt) to form a colored compound detected by spectrophotometry (570 nm). The amount of lipid hydroperoxides was measured by a microplate reader Synergie H1 (Biotech Labs, USA) and calculated from the standard curve of t-BuOOH. Results are expressed as nmol of peroxides per mg of proteins.”
- Protein carbonyl measurement:
“Protein carbonylation was evaluated as described by Chaouni et al. [7] in erythrocyte lysates and organ homogenates using a protein carbonyl content assay kit according to the manufacturer’s recommendations. Samples were derived in dinitrophenyl (DNP) hydrazone adducts by 2,4-dinitrophenylhydrazine (DNPH). Trichloroacetic acid was added to precipitate proteins. After an acetone washing step to remove excess DNPH and retain only proteins, a centrifugation was performed and pellets were suspended in a guanidine solution (6 M). The amount of protein carbonyls was measured at 375 nm by a microplate reader Synergie H1 (Biotech Labs, USA) and calculated from equation below. Results are expressed as nmol of protein carbonyls per mg of proteins.
|
C: amount of carbonyls in sample wells (nmol/well), P: amount of proteins from standard wells, D: dilution factor of samples, 1000: conversion factor (µg to mg).”
And finally, we explained RBE abbreviation line 72.
Reviewer 2 Report
The manuscript presents an interesting and useful set of experimental data on the effects of scattered and scanned proton beam irradiation on mice. The authors describe the results of thorough and carefully planned study. With the regard of presentation, I have the following comments:
- The authors used C57BL/6 mice. However, in the manuscript the strain is wrongly named C57Bl6. This should be changed.
- The legend to Figure 1 is very confusing. According to the text (page7, line 274) the survival curves do not significantly differ (p=0.5542). On the other hand, in the legend the authors mention the three probability cut-points (p<0.05, p<0.005, p<0.0001). If there is no significant difference in survival, what does it mean? The legend should be edited.
Author Response
We thank the reviewer for helpful comments.
- Indeed, the mouse strain is wrongly named. We changed C57Bl6 in C57BL/6 everywhere in the manuscript (every changes can be seen as we used the "Track Changes" mode).
- There is no difference in survival curves so we deleted “# for p < 0.05, ## for p < 0.005 and ### for p < 0.0001 for DS compared to PBS” in the legend of the Figure 1. In this way, results are more clear.